# Knockdown of Esr1 from DRD1-Rich Brain Regions Affects Adipose Tissue Metabolism: Potential Crosstalk between Nucleus Accumbens and Adipose Tissue

**DOI:** 10.3390/ijms25116130

**Published:** 2024-06-01

**Authors:** Dusti Shay, Rebecca Welly, Jiude Mao, Jessica Kinkade, Joshua K. Brown, Cheryl S. Rosenfeld, Victoria J. Vieira-Potter

**Affiliations:** 1Department of Nutrition and Exercise Physiology, Division of Food, Nutrition and Exercise Sciences, CAFNR, University of Missouri, Columbia, MO 65211, USA; dae7b6@mail.missouri.edu (D.S.);; 2Biomedical Sciences, University of Missouri, E102 Veterinary Medicine Building, Columbia, MO 65211, USArosenfeldc@missouri.edu (C.S.R.); 3MU Institute of Data Science and Informatics, University of Missouri, E102 Veterinary Medicine Building, Columbia, MO 65211, USA; 4Genetics Area Program, University of Missouri, E102 Veterinary Medicine Building, Columbia, MO 65211, USA; 5Thompson Center for Autism and Neurobehavioral Disorders, University of Missouri, E102 Veterinary Medicine Building, Columbia, MO 65211, USA

**Keywords:** midbrain region, reward pathway, physical activity, metabolism, adipose tissue, brown adipose tissue, estrogen, menopause

## Abstract

Declining estrogen (E2) leads to physical inactivity and adipose tissue (AT) dysfunction. Mechanisms are not fully understood, but E2’s effects on dopamine (DA) activity in the nucleus accumbens (NAc) brain region may mediate changes in mood and voluntary physical activity (PA). Our prior work revealed that loss of E2 robustly affected NAc DA-related gene expression, and the pattern correlated with sedentary behavior and visceral fat. The current study used a new transgenic mouse model (D1ERKO) to determine whether the abolishment of E2 receptor alpha (ERα) signaling within DA-rich brain regions affects PA and AT metabolism. Adult male and female wild-type (WT) and D1ERKO (KD) mice were assessed for body composition, energy intake (EE), spontaneous PA (SPA), and energy expenditure (EE); underwent glucose tolerance testing; and were assessed for blood biochemistry. Perigonadal white AT (PGAT), brown AT (BAT), and NAc brain regions were assessed for genes and proteins associated with DA, E2 signaling, and metabolism; AT sections were also assessed for uncoupling protein (UCP1). KD mice had greater lean mass and EE (genotype effects) and a visible change in BAT phenotype characterized by increased UCP1 staining and lipid depletion, an effect seen only among females. Female KD had higher NAc *Oprm1* transcript levels and greater PGAT UCP1. This group tended to have improved glucose tolerance (*p* = 0.07). NAc suppression of *Esr1* does not appear to affect PA, yet it may directly affect metabolism. This work may lead to novel targets to improve metabolic dysfunction following E2 loss, possibly by targeting the NAc.

## 1. Introduction

Loss of estrogen (i.e., 17β-estradiol, E2) in animals [1] and in humans is associated with reduced spontaneous and motivated physical activity (PA) [2] and dysfunctional adipose tissue metabolism [3], which together contribute to greater cardiometabolic risk following menopause [4]. Our work [5] and that of others [6] has shown that E2 affects adipose tissue via signaling through its steroid transcription factor receptors, ESR1 and ESR2 (ERα, ERβ), both of which are abundantly expressed in adipose tissues and brain. Regarding receptor-specific effects, the system-wide deletion of ERα causes obesity, insulin resistance, and adipose tissue dysfunction [7], whereas the system-wide deletion of ERβ causes a less robust phenotype but has recently been shown to adversely affect adipocyte metabolism [8].

There are important connections between the adipose tissue and the brain, and estrogen-induced effects on metabolism are likely mediated through both tissues. The brain affects adipose tissue metabolism in two important ways: First, sympathetic nervous system activation increases adipocyte mitochondrial activity and UCP1 expression (e.g., “browning”) via norepinephrine release, which binds to receptors on adipocytes [9]. Secondly, the brain mediates the ability of adipose tissue to regulate energy balance via the hormone leptin, which is released from adipose tissue and binds to receptors in specific brain regions to stimulate feelings of satiety and increase energy expenditure [10]. While many brain regions have been shown to affect adipose tissue, no studies to date have assessed how E2 signaling in the nucleus accumbens (NAc), the brain’s reward center, affects adipose tissue and systemic metabolism.

While little is known, the ability of E2 to modulate PA is possibly mediated through this brain region. Indeed, PA is a rewarding behavior, especially in rodents. We previously discovered that ovariectomy (i.e., ablation of all ovarian-derived hormones, including E2) in rats caused significant changes in dopamine (DA) receptor expression in the NAc, which were strongly associated with a suppression in voluntary wheel running [1]; similar relationships between running behavior and DA signaling in the NAc brain region have been reported [11]. It is well known that E2 affects DA synthesis and signaling [12,13,14], a relationship supported by our previous study, which demonstrated transcriptome differences in the NAc brain region in aromatase knock-out (ArKO) and wild-type (WT) control male and female mice [15]. We found that DA signaling and adipocyte lipid metabolism were among the most highly affected pathways in the ArKO mice, which was true in both sexes. In fact, the number one gene that was affected by E2 synthesis ablation was 6-pyruvolytetrahydropterin synthase (*Pts*), a gene encoding an enzyme necessary for catecholamine (e.g., DA) synthesis. Later, we learned that the suppression of *Pts* in mice causes excess visceral adiposity [16].

Hence, the purpose of this investigation was to investigate how *Esr1* knock-down on the DA receptor (DRD1) promotor (i.e., to target DA-rich brain regions, including the NAc) affects PA as well as systemic and adipose tissue metabolism. To this end, a novel mouse model was created that has suppressed E2/ERα signaling in the NAc brain region, the area of the brain with the highest density of the neurotransmitter DA. Here, the metabolic and behavioral phenotypes of this novel model are described. The hypothesis was that a unified mechanism involving the loss of signaling through ERα in DA-rich brain regions may help explain the wide range of behavioral and metabolic pathologies that occur in response to the loss of E2 signaling within the NAc region.

## 2. Results

### 2.1. Animal Model Confirmation

The model was created by crossing the commercially available DRD1-cre expressing mouse on the C57BL6 background with the *Esr1*-floxed mouse, also on the C57BL6 background. We validated the model by confirming genotype PCR, demonstrating that each mutated mouse tested positive for both Cre+ alleles, whereas control mice were heterogeneous for Cre. We named this mouse model “D1ERKO”, but it is a model of *Esr1* knock-down in DA-rich brain regions rather than a true knock-out. Thus, we refer to the D1ERKO as “KD” and Cre-mice as wild type (WT).

### 2.2. Effects of DRD1-Specific Esr1 Deletion on Body Weight and Body Composition

Neuronal estrogen signaling is known to affect body composition, but no prior studies have assessed how the down-regulation of *Esr1* in DA-rich brain regions per se affects body composition. Thus, we first characterized how the genetic mutation affected body composition. As shown in Figure 1A–C, D1ERKO were heavier (30.4 +/− 1.3 g vs. 27.4 +/− 0.8 g body weight) with more lean mass (22.4 +/− 0.5 g vs. 20.7 +/− 0.4 g lean mass). However, body fat percentage did not differ between genotypes, and there were no differences in fat pad weights between genotypes (Figure 1D). Circulating levels of leptin were also not different between genotypes (Figure 1E). Overall, males were predictably larger, with greater lean mass and fat mass, higher body fat percentage, and greater leptin levels compared to females of comparable genotypes. Females tended to have smaller fat pads, yet this effect only reached statistical significance for subcutaneous white adipose tissue (WAT).

### 2.3. Effects of DRD1-Specific Esr1 Deletion on Energy Intake and Expenditure

In order to gain insight into the etiology of any differences in body mass and composition, we assessed energy intake and expenditure. We were particularly interested in how the brain-specific manipulation of *Esr1* affected both resting energy expenditure and PA. As shown, KD mice consumed (Figure 2A) and expended (Figure 2C) more total energy; however, these genotype differences appeared to be due to their greater lean mass because these differences were not significant after adjusting for body mass (Figure 2B,D). There were no differences between genotypes in spontaneous PA (SPA) (Figure 2E). Expected sex differences occurred, which were not affected by genotype: females consumed less total energy but consumed and expended more energy relative to their size.

### 2.4. Effects of DRD1-Specific Esr1 Deletion on Brown Adipose Tissue Phenotype

Given the trends toward greater intake and expenditure in the absence of any such trends toward increased PA, we assessed BAT activation (i.e., an indicator of adaptive thermogenesis) in KD vs. WT mice via assessment of BAT phenotype via histology and gene expression (Figure 3). As shown, there was a visible increase in BAT UCP1 staining in the KD mice of both sexes, but this genotype effect did not reach statistical significance, likely due to insufficient power to detect differences (main effect of genotype, *p* = 0.1; observed power for the corrected model, 0.263; n = 5/group), females also tended to have greater BAT UCP1 staining (*p* = 0.087) (Figure 3A). In addition, quite strikingly, and only among females, KD BAT was lipid-depleted, indicative of high BAT activity (Figure 3B), which relies on local fat oxidation. Lipid depletion was also indirectly assessed via BAT nuclei density, which also appeared to be greater among female KD, although this did not reach statistical significance, likely due to insufficient power to detect differences (genotype by sex interaction, *p* = 0.27; observed power for the corrected model, 0.219) (Figure 3C). Also, in support of the relationship between BAT activity and fat oxidation, BAT activity-related gene expression (i.e., *Ucp1*—uncoupling protein 1, *Adrb3*—beta 3 adrenergic receptor), indicative of uncoupling activity (which drives fat oxidation), correlated significantly and inversely with circulating NEFA levels. BAT gene expressions of *Ucp1* and *Adrb3* were not significantly different, yet for *Esr2* expression, there were significant effects on both sex (females higher) and genotype (KD higher); this genotype effect was much stronger among females, indicated by sex by genotype interaction. Females had greater levels of BAT *Esr1* expression (Figure 3D).

### 2.5. Effects of DRD1-Specific Esr1 Deletion on Insulin Sensitivity and Glucose Tolerance

Body composition and systemic metabolism are known to affect insulin sensitivity, and brain DA signaling, in particular, has been shown to be associated with insulin sensitivity. Thus, we assessed insulin sensitivity and glucose tolerance in D1ERKO and WT mice. Remarkably, despite no differences in total adiposity, female D1ERKO mice trended toward having improved glucose tolerance (lower GTT area under the curve (AUC), *p* = 0.07); this effect was not observed among males (Figure 4A,B). Similarly, there was a significant genotype x sex interaction for fasting insulin levels such that levels were lower among KO females but higher among KO males (Figure 4C); no differences were found for fasting NEFAs (Figure 4D). As expected, females exhibited greater glucose tolerance and lower fasting insulin than males.

### 2.6. Effects of DRD1-Specific Esr1 Deletion on White Adipose Tissue Browning

In addition to BAT activity contributing to increased energy expenditure and improved glucose metabolism, the “browning” of WAT has been shown to have similar metabolic effects [17,18,19]; like BAT activation, the browning of WAT is driven by catecholamine stimulation, which is mediated centrally. Cold, chemical, and exercise-mediated sympathetic nervous system stimulation is known to induce brain catecholamine synthesis (e.g., DA, norepinephrine); norepinephrine binds to beta 3 adrenergic receptors (B3ARs) on adipocytes, stimulating both BAT and browning (i.e., increased UCP1 expression and mitochondrial biogenesis/activity) of WAT [20]. Thus, we assessed WAT browning via the assessment of browning genes as well as histological analysis (Figure 5). There was a trend toward a main effect of genotype on WAT Ucp1 gene expression, which tended to be lower among KD (*p* = 0.05). There was also a significant sex-by-genotype interaction such that, among males only, WAT leptin mRNA levels were higher among KD mice compared to WT males. We and others have shown females to be more sensitive to WAT browning [21,22], a finding supported herein. Indeed, there were sex differences in WAT UCP1 staining, gene expression, and protein expression. There was a sex-by-genotype interaction for UCP1 protein expression, such that, among males only, KD had lower levels (0.19 ± 0.17 vs. 0.12 ± 0.14 relative expression for male WT vs. male KD, respectively). In addition, female WAT expressed greater gene expression levels of adiponectin (*Adipoq*) and lower levels of leptin (*Lept*). Females also had greater ESR1 and adiponectin protein levels in WAT, yet none of these effects were affected by genotype.

### 2.7. Effects of Sex and D1ERKO Genotype on Nucleus Accumbens (NAc) Gene Expression and Tyrosine Hydroxylase Staining

The following experiments were performed to determine how the genetic mutation affected DA signaling, E2 signaling, and DA content. Presented in Figure 6 are DA- and E2-related gene expression and tyrosine hydroxylase (TH) staining data from the NAc brain region. TH staining did not differ between sexes or genotypes (Figure 6A,B). In comparing sexes for NAc gene expression across the entire cohort (Figure 6C), *Oprm1* was higher among females (sex effect) and D1ERKO mice (genotype effect). No other genotype differences were found, yet among females only, D1ERKO had significantly greater levels of *Esr2*, and there was a sex-by-genotype interaction for *Per3* (i.e., period circadian regulator 3, a primary component of the circadian clock system). Among males, *Per3* was lower among D1ERKO.

### 2.8. Correlations among Nucleus Accumbens Gene Expression, Adipose Tissue Phenotype, Activity Levels, and Metabolism

To identify potential relationships among NAc gene expression, DA content, BAT and WAT phenotype, systemic metabolism, and PA, we performed correlation analyses among variables across the entire cohort of animals (Table 1). NAc DA content, as assessed via tyrosine hydroxylase (TH) content, correlated significantly with basal metabolism (resting energy expenditure relative to body weight (REE)/TH, r = 0.499), the BAT gene expression of *Ucp1* (r = 0.713), and the BAT gene expression of *Esr2* (r = 0.667). REE, in addition to correlating with NAc DA content, correlated with relative energy intake (r = 0.492), cage PA (r = 0.314), and WAT UCP1 protein content (r = 0.379) and staining intensity (r = 0.543). REE was associated inversely with adiposity (i.e., percent body fat, r = −0.406), visceral fat content (r = −0.302), WAT *Lept* expression (r = −0.375), and glucose tolerance (glucose AUC, r = −0.399). The NAc expression of the opioid receptor *Oprm1* correlated with NAc dopaminergic genes (DA transporter, *Slc6a3*: r = 0.56; *Pts*: r = 0.525) and the clock gene *Per3* (r = 0.537) and, notably, was strongly correlated with NAc gene expression of *Esr2* (r = 0.657). *Oprm1* was very strongly and inversely correlated with WAT *Lept* expression (r = −0.724) and moderately with body fat percentage (r = −0.477). Finally, the most remarkable and surprising finding was that NAc *Oprm1* gene expression was strongly correlated with improved glucose tolerance (r = 0.73). Glucose tolerance also was correlated with the DA transporter gene, NAc *Slc6a3* (r = 0.388), and NAc *Esr2* (r = 0.666). Better glucose tolerance was also associated with greater BAT *Esr2* (r = 0.358) and BAT *Esr1* expression (r = 0.385), as well as with ESR2 protein content in WAT (r = 0.388, *p* = 0.041). Expectedly, glucose tolerance was also associated with greater WAT UCP1 staining (r = 0.697) as well as UCP1 protein expression (r = 0.376). BAT UCP1 correlated with genes associated with E2 such as BAT *Cyp19a* (i.e., aromatase) (r = 0.849), as well as *Esr1* and *Esr2* (r = 0.61; r = 0.71). PGAT UCP1 staining and protein expression also correlated with REE, as stated above, as well as better glucose tolerance and lower overall adiposity; in addition, it correlated inversely with WAT *Lept* gene expression (r = −0.648).

## 3. Discussion

Loss of E2 adversely affects metabolic health, leading to increased visceral adiposity and enhanced risk for insulin resistance and cardiometabolic disease [4], yet the mechanisms are not fully understood. Loss of E2 has been shown in animal models to reduce voluntary PA, which exacerbates weight gain and metabolic dysfunction [23], a behavioral change that may be driven by changes in the NAc brain region [15]. Our prior work suggests that E2 loss is associated with dysregulated DA synthesis in the NAc brain region, the region responsible for driving motivated behaviors, such as wheel running [11]. To test the hypothesis that E2 signaling in the NAc brain region is permissive for voluntary PA, in the present study, we used a mouse model where a major form of the E2 receptor, E2 receptor alpha (*Esr1*), was down-regulated in DA-rich brain regions. We characterized this model (i.e., D1ERKO), compared to WT control mice in terms of spontaneous PA (SPA), as well as several indicators of systemic and adipose tissue-specific metabolism. Our prediction at the outset was that the ablation of *Esr1* selectively in the NAc region would reduce SPA and alter body composition and metabolism, possibly in a sex-specific manner. While the selective down-regulation of this gene within this brain region did not impact SPA, we did detect some differences related to adipose tissue and metabolic activity.

We previously demonstrated that selective breeding for high PA in rats associates with changes in DA receptor gene expression in the NAc brain region, whereas OVX significantly suppresses those same DA-related genes. Across all of the rats in that study, there was a strong correlation between DA-related genes and voluntary wheel running [1]. Whole-body *Esr1* deletion recapitulates the suppressed PA phenotype of OVX [24], suggesting that the mechanism may involve suppressed *Esr1* signaling in the brain. In comparing NAc brain regions from aromatase knock-out (ArKO) and wild type (WT) male and female mice [15] using RNAseq as a way to investigate how E2 affects this brain region, ArKO demonstrated a signature of abnormal NAc-specific DA signaling pathways, with the number one down-regulated gene being *Pts*, a gene responsible for TH activity (i.e., the enzyme necessary for DA synthesis). The two pathways most robustly affected by the lack of E2 were those associated with DA signaling and adipose tissue metabolism. Those data led us to create the D1ERKO model in order to test the hypothesis that E2 signaling via *Esr1* in the NAc may affect voluntary PA and adipose tissue metabolism.

The DRD1-specific ablation of *Esr1* did not affect NAc DA receptor gene expression or DA synthesis (i.e., TH activity). Thus, our model did not allow us to directly address whether E2-mediated DA synthesis affects PA, yet the lack of differences in PA between genotypes does not necessarily disprove the hypothesis that the suppression of DA mechanistically explains why E2 loss leads to physical inactivity. D1ERKO mice did, however, have greater NAc gene expression levels of mu-opioid receptor (*Oprm1*), supporting that even modest changes in E2 signaling affect DA-related pathways. Prior studies have shown *Oprm1*, which is associated with reward-driven feeding, to be modulated by E2 [25]. We confirmed this in our prior RNAseq study, where we showed *Oprm1* to be among the top genes down-regulated with E2 loss [15]. In the current study, we showed that NAc *Oprm1* was strongly associated with visceral adiposity (inverse) and the expression of WAT UCP1 (i.e., WAT browning) (positive). This gene was also more highly expressed among females, independent of genotype, and increased in the KD. Most strikingly, NAc *Oprm1* correlated very strongly with improved glucose tolerance.

D1ERKO mice weighed more and had greater lean mass. There was also a small increase in energy expenditure despite no change in PA. BAT thermogenesis, driven by UCP1, contributes to resting energy expenditure. Given the known sex differences in BAT [22] and WAT browning [21,26] and accumulating evidence that central E2 signaling affects BAT activation [27], it is not entirely surprising that BAT appeared to be affected by the *Esr1* mutation in the present study. It is well established that E2 signaling is permissive for WAT browning, which may be mediated centrally [28], although the mechanisms are not clear. We previously established that ERα is not required for WAT browning; moreover, those ERα knock-out mice (presumably with unrestricted ERβ signaling) were more responsive to browning [18]. It is now known that systemic ERβ activation causes WAT browning [29], and we showed that ERb is required for exercise-induced browning [30] and that ERβ is induced by stimuli that induce browning [21]. Regarding the BAT phenotype observed in the present study, a robust increase in BAT ERβ expression (i.e., *Esr2*) in the female D1ERKO mice coincided with indicators of BAT activity, including phenotypic evidence, and increased the expression of UCP1, as well as with increased non-exercise energy expenditure. How the DRD1-specific down-regulation of *Esr1* would affect BAT E2 receptors is uncertain. We hypothesize that increased signaling through ERβ, due to the increased relative presence of this receptor, may have affected adipose tissue via centrally mediated browning, mediated by DA production and its subsequent conversion to norepinephrine (Figure 7). While NAc gene levels of *Esr2* were not different between genotypes, the strong correlations between NAc *Esr2* expression and indices of BAT and WAT browning support the hypothesis that signaling through neuronal *Esr2* may improve adipose tissue metabolism. We hypothesize that the phenotype observed in the D1ERKO mice may be attributed not solely to the suppression of *Esr1* but rather to increased *Esr2* signaling (Figure 7). Due to much overlap in functions mediated through ERα and ERβ (e.g., regulation of DA metabolism, which appears more attributed to *Esr2* vs. *Esr1* [12,31]), it is possible that compensatory signaling through ERβ may have prevented more robust genotype differences from being observed.

ERb, while not influencing PA [32], is the major E2 receptor expressed in the brain [33] and has recently been shown (using various ERb-selective ligands administered systemically) to improve metabolism, in part by inducing the browning of WAT [29,34]. No studies have yet directly tested the effect of neuronal ERb on adiposity or metabolism, but one recent study found that deletion of a tumor suppressor unexpectedly promoted leanness in female (but not male) mice, an effect that coincided with increased neuronal proliferation and homeostasis that was driven via ERb [35], making a strong case for the further exploration of the influence of neuronal ERb on metabolism. It is likely that E2’s effects on adipose tissue browning/mitochondrial metabolism are at least partially mediated through its actions in the brain, where ERs are heavily expressed and where effects are largely non-genomic [36]. Our demonstration of sexual dimorphism further supports the hypothesis of the critical involvement of ERb signaling; the unexpected metabolic “benefits” were specific to the female mice, and those systemic benefits coincided with the augmentation of ERb, both in NAc and BAT.

Another sexually dimorphic finding warrants highlighting—the NAc expression of *Per3* was affected in a sexually divergent manner, with KD females experiencing an increase and KD males experiencing a decrease. *Per3* is a circadian regulator known to be affected by DA [37]. It was also one of the top NAc-specific genes identified as being regulated by E2 in our prior study [15]. Aggarwal and colleagues found that the ablation of *Per3* caused an increase in adipogenesis, supporting the hypothesis that central *Per3* signaling affects adipose tissue metabolism [38]. In our study, this is supported in that male KD mice had both lower NAc *Per3* and higher WAT leptin levels. And, the opposite was true in females: KD mice had higher NAc *Per3* and an improved WAT phenotype. Further study is required to determine the role that *Per3* may play in sex differences in the centrally mediated effects of E2.

### Study Limitations

The DRD1 promotor was chosen as the ideal means to selectively target *Esr1* in the NAc brain region, given the high density of DA receptors in this brain region, with DRD1 being the predominant receptor. However, DA receptors are expressed in several other brain regions as well and may also be expressed at lower levels throughout the body. Thus, our model was not a full NAc-specific *Esr1* deletion but rather a knock-down. Moreover, likely because *Esr1* is expressed on so many cell types in the brain, DRD1-specific deletion did not result in a detectable difference at the mRNA level in total *Esr1* expression in NAc between KD and WT mice. Future studies should consider using DREADD to more precisely target ERs in the NAc. Given the size of the brain region, we were not able to perform histology and assess gene and protein expression; thus, there are no protein expression data to report. Nonetheless, each KD animal was validated via genotypic confirmation (i.e., all mice expressed flox sites within the *Esr1* gene and were crossed with mice heterogeneously expressing Cre recombinase enzyme on the DRD1 promoter; homozygous Cre+ mice were considered KD). Thus, the genotype differences observed may only be attributed to the genetic effect imposed and confirmed (the expression of Cre recombinase as well as the floxed *Esr1* gene on the DRD1 promoter). In addition, an outcome reported herein was the browning of WAT and activation of BAT, both of which would have required a cold or chemical stimulus to activate. Thus, we only assessed constitutive BAT and WAT activity and did this only indirectly via UCP1 expression—future studies should assess BAT and WAT mitochondrial function directly (e.g., fuel-driven oxygen consumption via oroboros or Seahorse instruments) both constitutively and in response to cold or exercise stimulation. Future studies should also assess voluntary wheel running as that is a much better assessment of motivation for physical activity, along with other motivated behaviors. Finally, we were not able to measure circulating levels of sex hormones, so it is unknown whether the genetic mutation affected sex hormone production, but this is unlikely given the specificity of the *Esr1* deletion.

## 4. Materials and Methods

*Mouse model and basic study design:* In order to generate a mouse model with significantly reduced expression of ESR1 in DRD1-rich brain regions, DRD1-Cre+ mice (Jackson Laboratory, #037156-JAX) were crossed with homozygous ESR1-floxed mice (in house) bred on a C57Bl6/J background. Male and female mice of both genotypes (n = 17–22/group) were compared. Following in vivo assessments, animals were humanely euthanized in accordance with the American Veterinary Medical Association guidelines for euthanasia of animals. Upon sacrifice, whole brain, fasted blood plasma, liver, interscapular brown adipose tissue (BAT), subcutaneous/inguinal (SQAT), and visceral/perigonadal white adipose tissue (PGAT) were extracted, and tissue weights were collected and analyzed, as described below. 

*Diet and housing conditions:* Wild-type (WT) (i.e., heterozygous for expression of Cre recombinase) and Esr1-DRD1 specific knock-down (D1ERKO) (i.e., homozygous for Cre) mice were bred and housed at University of Missouri Animal Sciences Research Center (Columbia, MO, USA) under a normal 12-12 h light–dark cycle with room temperature maintained at 21.7C (71F) with food (phytoestrogen-free AIN93G diet, Bio-Serve, Flemington, NJ, USA) and water available ad libitum. All mice were weaned at 21 days and housed in pairs or singly based on sex and genotype. Approximately two weeks prior to sacrifice, animals were assessed for in vivo behavioral and metabolic experiments, as described below; they were all sacrificed, and tissues were collected at ~23 weeks of age. At the end of the study, animals were euthanized via CO_2_ inhalation followed by cervical dislocation, and tissues were harvested and either fixed (10% formalin, adipose, or 4% paraformaldehyde, brain) or snap-frozen in liquid nitrogen and stored at −80 °C until analyses.

*Assessment of energy intake and expenditure:* All animals were assessed weekly for energy intake (EI) and body weight. Resting and total energy expenditure (EE) and spontaneous physical activity (SPA) were assessed via metabolic chambers (72 h observation period) utilizing indirect calorimetry and 3-plane beam break detection (Promethion 8-cage system, Sable Systems, Las Vegas, NV, USA). Data from the first 24 h period was not used as this time was considered the habituation period. The 48 h run captured at least two light and two dark cycles.

*Body composition:* Percent body fat (BF%) and lean mass were measured by a nuclear magnetic resonance imaging whole-body composition analyzer (EchoMRI 4in1/1100; Echo Medical Systems, Houston, TX, USA) 24–48 h prior to assessment in the metabolic chambers.

*Brain collection and preparation:* The whole brain was collected and snap-frozen in liquid nitrogen and stored at −80 °C until analysis. Whole brains were rapidly sectioned on dry ice, and tissue punches of the NAc region were snap-frozen in liquid nitrogen. This brain region was located and dissected using the Allen mouse brain atlas [39]. Brain slices were taken using a Zivic brain block (Adult mouse brain slicer matrix with 1.0 mm coronal section slice intervals; Zivic Instruments, Pittsburgh, PA, USA) as a guide. Micro punch samples were taken of the NAc using a 2 mm Harris micro punch (GE Healthcare Bio-sciences, Marlborough, MA, USA). Bilateral NAc punches were combined and snap-frozen in liquid nitrogen and stored at −80 °C until further analyses.

*Glucose tolerance:* An intraperitoneal glucose tolerance test (GTT) was performed as previously described [24]. Briefly, following a six-hour fast, baseline blood (time 0) was collected from a nick in the tail vein and sampled by a hand-held glucometer (AlphaTRAK; Abott Laboratories, Abbott Park, IL, USA). Then, an intraperitoneal injection of a sterile solution of 50% dextrose (2 g/kg BW) was administered. Blood glucose measurements were taken at 15, 30, 45, 60, and 120 min following injection. The glucose area under the curve (AUC) above the baseline was calculated as a measure of glucose tolerance.

*Blood biochemistry:* Fasted-state circulating levels of leptin, insulin, and non-esterified fatty acids (NEFAs) were measured using commercial kits (insulin: Crystal Chem, Elk Grove Village, IL, USA, #90080; leptin: Crystal Chem, #90030; NEFAs: ZenBio, Research Triangle Park, NC, USA; Kit: SFA-5). All samples were run in duplicate with acceptable coefficients of variation.

*Real-time quantitative PCR (rtPCR):* Following RNA extraction (RNeasy Lipid Tissue Kit, Qiagen, Venlo, Netherlands), total RNA was assayed using a Nanodrop spectrophotometer (Thermo Scientific, Wilmington, DE, USA) to assess purity and concentration. All RNA samples had 260/280 readings between 1.9 and 2.1, indicating excellent quality and purity. First-strand cDNA was synthesized from total RNA using the High Capacity cDNA Reverse Transcription kit (Applied Biosystems, Carlsbad, CA, USA). Quantitative real-time PCR was performed using the ABI StepOne Plus sequence detection system (Applied Biosystems). Primer sequences were designed using the NCBI Primer Design tool. All primers were purchased from Sigma-Aldrich (St. Louis, MO, USA) or Integrated DNA Technologies (Carolville, IA, USA). The internal housekeeping control gene (HKG) used was 18 s, and the cycle threshold (CT) was not different among the groups of animals. mRNA expression is expressed as 2^∆CT^ where ∆CT = HKG CT—gene of interest CT. Primer sequences are provided in Table 2.

*Adipose tissue histology:* Formalin-fixed samples were processed through paraffin embedment, sectioned at 5 µm in thickness (interscapular BAT, visceral WAT (PGAT depot), and stained in a 1:1200 dilution with UCP1 antibody for 30 min with a heat-induced epitope retrieval (HIER) pretreatment, using DAKO brand citrate in a decloaking chamber. Sections were evaluated via an Olympus BX34 photomicroscope (Olympus, Melville, NY, USA), and images were taken via an Olympus SC30 Optical Microscope Accessory CMOS color camera. An investigator blinded to the groups performed all procedures.

*NAc tyrosine hydroxylase staining:* To assess DA synthesis, we performed immunofluorescence staining for tyrosine hydroxylase (TH) in the NAc brain regions of male and female mice. Following sac, which was immediately followed by cervical dislocation, whole brains were collected fresh and immediately placed in 4% paraformaldehyde for 24–48 h. Following this, brains were transferred to a 15 mL conical tube containing 30% sucrose solution and allowed to sit until the brain sank to the bottom of the tube, indicating that water had been sufficiently displaced by the sucrose in order to minimize tissue damage during freezing. Brains were sectioned at 20 μm using a vibratome VT1000S (Leica Biosystemis, Nussloch, Germany) to visualize the NAc. Sections were stored in a cryoprotective solution containing 0.1 M sodium phosphate buffer (PBS, pH 7.2), ethylene glycol, sucrose, and polyvinylpyrrolidone at −20 °C until ready for staining. Tissue sections were washed with 0.01 M phosphate-buffered saline 3 times for 10 min each at room temperature to remove cryoprotectant. Sections were pre-blocked in PBS-0.3% Triton X100 (PBS-T) containing 10% normal donkey serum (Millipore, Saint Louis, MO, USA, catalog # S30) for 30 min at room temperature. Sections were then incubated in rabbit-anti tyrosine hydroxylase antibody (TH, 1:2000, Chemicon (Rolling Meadows, IL, USA) #AB152) in PBS-T with 1% normal donkey serum for 24 h at 4 °C. Following 0.01 M PBS wash (5 × 10 min), sections were incubated at room temperature for 2 h in donkey anti-rabbit serum conjugated to Cy3 (1:200, Jackson Immuno Research, West Grove, PA, USA, #11-165-152), then coverslipped with Prolong Diamond antifade with DAPI (Invitrogen, Carlsbad, CA, USA, cat P36962). Control sections were incubated without primary antibodies. An Olympus epifluorescent spinning disk confocal system equipped with an Orca Hamamatsu CCD camera was used to examine sections.

*Statistics:* Main effects of genotype and sex, as well as genotype x sex interactions, were determined using 2-way ANOVA. When significant interactions occurred within sex, 1-way ANOVA was performed to determine genotype differences; to determine sex differences within genotype, 1-way ANOVA was performed. All data are presented as mean ± SEM; n = 8–11/group, and *p* values ≤ 0.05 were considered statistically significant. Analyses were performed using SPSS version 25 (IBM, Armonk, NY, USA). Pearson correlations were also calculated among the outcome variables. A strong correlation was defined as having an R-value > ±0.7, a moderate correlation having an R-value of ±0.5–0.7, and a weak correlation having an R-value < ±0.5.

All animal experiments described herein were approved by the Institutional Animal Care and Use Committee (IACUC #9474, approval date 11/9/2018) of the University of Missouri.

## 5. Conclusions

Selective genetic modification of *Esr1* in DA-rich brain regions, despite not affecting SPA or total adiposity, was shown to affect systemic and adipose tissue metabolism in a sex-specific way. This mutation led to sex-divergent changes in metabolism—among females, there was a trend toward improved glucose tolerance; among males, there was an increase in WAT *Lept* (often indicative of leptin resistance). These sex interactions paralleled changes in NAc genes, most notably *Per3*, a circadian regulator. Collectively, these findings reveal novel relationships between E2 signaling in DA-rich brain regions, BAT activity, and glucose homeostasis.

## Figures and Tables

**Figure 1 ijms-25-06130-f001:**
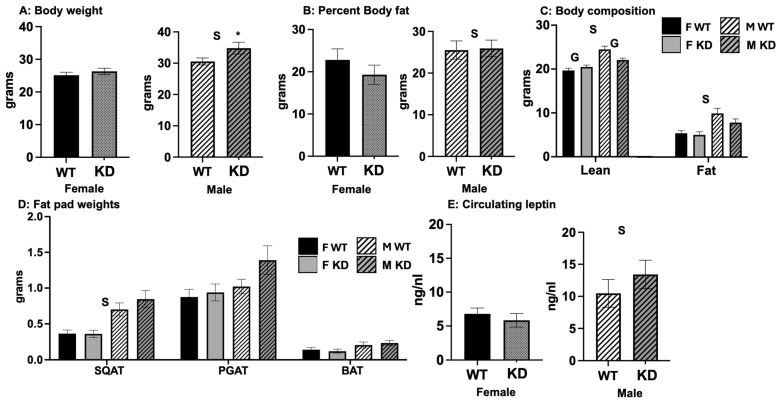
*Body weight and composition.* (**A**). Body weight at end of study; (**B**). body fat percentage; (**C**). fat mass and lean mass; (**D**). fat pad weights. (**E**) circulating weight. Values expressed as Means ± SEM (n = 16–22/group). WT = Wild-type; KD = Knock-down; F = Female; M = Male; SQAT = subcutaneous white adipose tissue; PGAT = perigonadal white adipose tissue; BAT = brown adipose tissue. S, main effect of sex, *p* < 0.05; G main effect of genotype, *p* < 0.05; * genotype difference within sex, *p* < 0.05, yet no main effect of genotype and no significant interaction.

**Figure 2 ijms-25-06130-f002:**
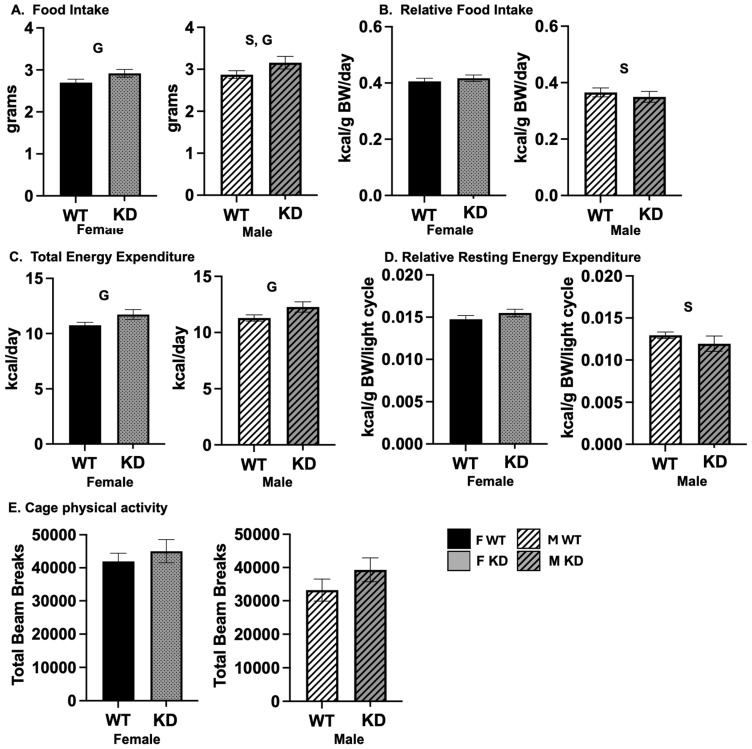
*Food intake and energy expenditure.* (**A**). Food intake; (**B**). food intake relative to body weight; (**C**). total energy expenditure; (**D**). resting energy expenditure relative to body weight; (**E**). cage physical activity. Values expressed as Means ± SEM (n = 16–22/group). WT = Wild-type; KD = Knock-down; F = Female; M = Male; S, main effect of sex, *p* < 0.05; G main effect of genotype, *p* < 0.05.

**Figure 3 ijms-25-06130-f003:**
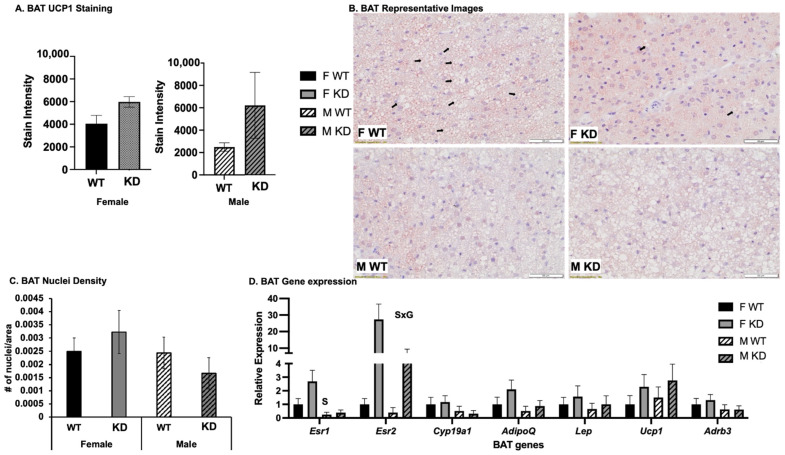
*BAT phenotype assessment.* (**A**). BAT UCP1 staining intensity (n = 5/group); (**B**). BAT representative images; (**C**). BAT nuclei density as an indirect indicator of lipid depletion (n = 5/group); (**D**). BAT mRNA expression expressed as 2^dct^, where dct = gene of interest—normalizer gene (*Bactin*). Data are shown as means ± SEM (n = 13–16/group) and expressed relative to F WT. WT = Wild-type; KD = Knock-down; F = Female; M = Male. S, main effect of sex, *p* < 0.05; G, main effect of genotype, *p* < 0.05; SxG interaction, *p* < 0.05. Arrows indicate areas of high lipid droplet density, which is depleted in female KD; among males, lipid droplet profile was similar among genotypes, and thus, arrows are not used to denote major differences; images are at 40× magnification; scale bar showing 50 μm.

**Figure 4 ijms-25-06130-f004:**
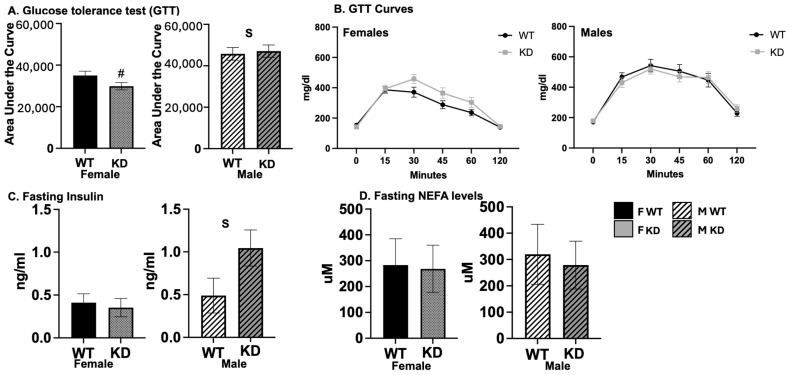
*Glucose tolerance.* (**A**). Glucose area under the curve for females (left) and males (right); (**B**). glucose values throughout test for females (left) and males (right); (**C**). fasting insulin values; (**D**). fasting NEFA levels; values expressed as means ± SEM (n = 13–16/group). WT = Wild-type; KD = Knock-down; F = Female; M = Male; NEFA = non-esterified fatty acids. S, main effect of sex, *p* < 0.05; # trend for genotype difference within sex, *p* = 0.07 (non-significant trend).

**Figure 5 ijms-25-06130-f005:**
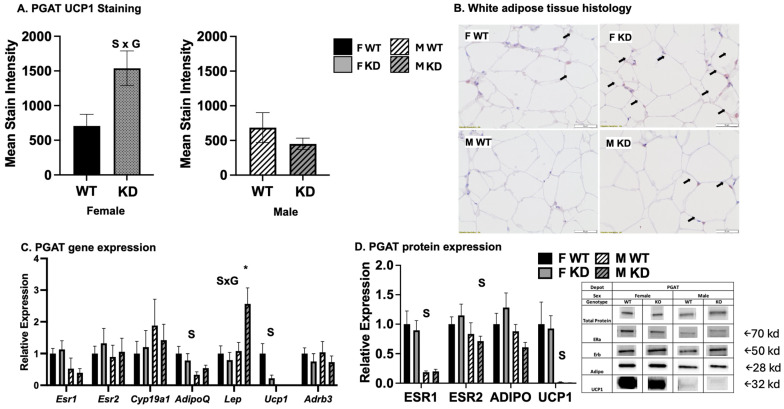
*White adipose tissue browning assessment.* (**A**). UCP1 stain intensity (n = 3–5/group); (**B**). WAT representative UCP1-stained images; (**C**). white adipose tissue (WAT) gene expression from the perigonadal (PGAT) depot); * *p* < 0.05 compared to all groups based on post-hoc Tuke’s test. Values expressed as means ± SEM (n = 13–16/group). WT = Wild-type; KD = Knock-down; F = Female; M = Male; S, main effect of sex, *p* < 0.05; G main effect of genotype, *p* < 0.05; S x G, interaction, *p* < 0.05. WAT perigonadal (PGAT) depot mRNA expression expressed as 2^dct^, where dct = gene of interest—normalizer gene (beta actin) and shown relative to F WT; data reported as means ± SEM (n = 13–16/group); (**D**). Relative UCP1, adiponectin, ERα, and ERβ protein expression expressed relative to total normalizer protein, beta tubulin (n = 8/group). Black arrows indicate positive UCP1 staining.

**Figure 6 ijms-25-06130-f006:**
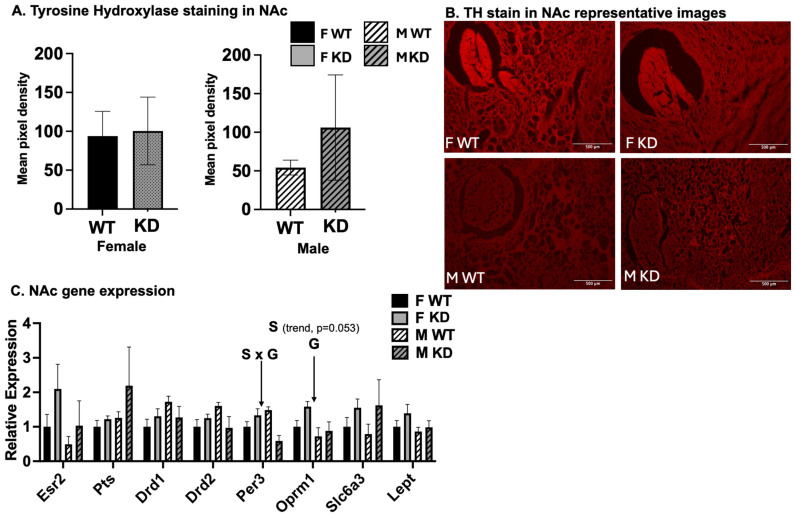
*Nucleus accumbens phenotype.* (**A**). NAc tyrosine hydroxylase staining. Values expressed as means ± SEM (n = 13–16/group); mRNA expression expressed as 2^dct^, where dct = gene of interest—normalizer gene (beta actin) and expressed relative to F WT; data presented as means ± SEM; (**B**). representative images (400× magnification; scale bar showing 500 μm) of TH staining in NAc; (**C**). NAc gene expression. WT = Wild-type; KD = Knock-down; F = Female; M = Male; S, main effect of sex, *p* < 0.05; G main effect of genotype, *p* < 0.05.

**Figure 7 ijms-25-06130-f007:**
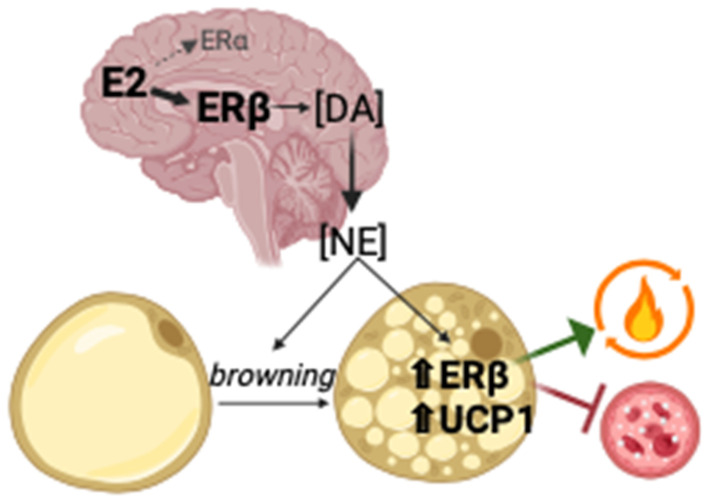
Hypothetical model by which suppression of *Esr1* in DA-rich brain regions leads to increased brain ERβ signaling, which increases DA synthesis, ultimately improveing metabolic health. DA’s conversion to NE lead to incrased BAT activity and WAT browning, demonstrated by UCP1 expression. Activaiton of BAT and browning of WAT increases ERβ expression, which facilitates mitochondrial fat oxidation and improves glucose metabolism. E2 = 17b-Estradiol; ER = estrogen receptor; DA = dopamine; NE = norepinephrine; UCP1 = uncoupling protein 1.

**Table 1 ijms-25-06130-t001:** Correlation matrix.

	Lean mass (rel BW)	PGAT (g)	BAT (g)	GTT AUC	SPA (daily)	REE (relBW)	INSULIN [ng/mL]	LEPTIN [ng/mL]	NEFA [μM]	TH stain	NAc *Oprm1*	NAc *Per3*	NAc *Pts*	NAc *Slc6a3*	NAc *Esr2*	BAT *Ucp1*	BAT *Adrb3*	BAT *Esr1*	BAT *Esr2*	BAT *Adipoq*	BAT *Lept*	PGAT *Ucp1*	PGAT *Adrb3*	PGAT *Esr2*	PGAT *Adipoq*	PGAT *Lept*	BAT UCP1 stain	PGAT UCP1 stain	PGAT ESR1	PGAT ESR2	PGAT UCP1	PGAT ADIPOQ
**Lean mass** **(rel BW)**	1	−0.823 **	−0.437 **	−0.600 **	0.053	0.375 **	−0.644 **	−0.829 **	−0.1	0.355	0.431 *	−0.01	0.041	0.383 *	0.354	−0.144	0.132	0.101	0.203	0.138	0.157	−0.01	0.189	0.231	0.154	−0.528 **	0.31	0.38	0.451 *	0.547 **	0.356 *	0.176
**PGAT (g)**	−0.823 **	1	0.458 **	0.514 **	0.016	−0.302 **	0.680 **	0.854 **	0.124	−0.19	−0.208	0.051	−0.234	−0.228	−0.124	0.091	−0.183	−0.144	−0.199	−0.148	−0.207	−0.046	−0.196	−0.258	−0.161	0.652 **	−0.204	−0.382	−0.285	−0.400 *	−0.191	−0.31
**BAT (g)**	−0.437 **	0.458 **	1	0.433 **	0.013	−0.257 *	0.601 **	0.570 **	0.025	−0.415	−0.254	−0.163	−0.437 *	−0.17	0.139	0.023	−0.198	−0.235	−0.213	−0.21	−0.232	−0.153	−0.232	−0.264	−0.236	0.471 **	−0.246	−0.423	−0.435 *	−0.356	−0.421 *	−0.343
**GTT AUC**	−0.600 **	0.514 **	0.433 **	1	−0.258	−0.399 **	0.322 *	0.472 **	−0.014	−0.181	−0.730 **	−0.425	−0.091	−0.594 *	−0.666 *	0.098	−0.066	−0.102	−0.246	−0.01	−0.207	−0.051	0.146	−0.052	−0.084	0.543 **	−0.318	−0.697 **	−0.385 *	−0.388 *	−0.376 *	−0.331
**SPA (daily)**	0.053	0.016	0.013	−0.258	1	0.314 **	0.141	0.072	−0.031	0.227	0.131	0.323	−0.103	0.085	−0.046	0.097	0.066	0.065	0.166	0.151	0.325 *	−0.024	−0.215	−0.233	−0.175	−0.251	0.16	0.122	0.467 **	0.448 *	0.314	0.3
**REE (relBW)**	0.375 **	−0.302 **	−0.257 *	−0.399 **	0.314 **	1	−0.272 *	−0.410 **	0	0.498 *	0.254	0.141	−0.125	0.262	0.057	0.069	0.141	0.202	0.244	0.189	0.197	0.144	0.024	−0.077	−0.044	−0.375 **	0.236	0.543 *	0.324	0.351	0.379 *	0.28
**INSULIN [ng/mL]**	−0.644 **	0.680 **	0.601 **	0.322 *	0.141	−0.272 *	1	0.806 **	0.237	−0.267	−0.286	−0.215	−0.249	−0.22	−0.235	−0.062	−0.284 *	−0.226	−0.204	−0.238	−0.315 *	−0.087	−0.243	−0.249	−0.279 *	0.439 **	0.022	−0.332	−0.333	−0.35	−0.236	−0.376 *
**LEPTIN [ng/mL]**	−0.829 **	0.854 **	0.570 **	0.472 **	0.072	−0.410 **	0.806 **	1	0.19	−0.336	−0.331	−0.009	−0.066	−0.303	−0.305	−0.047	−0.264	−0.256	−0.275 *	−0.264	−0.317 *	−0.099	−0.266	−0.319 *	−0.218	0.521 **	−0.118	−0.540 *	−0.396 *	−0.377 *	−0.343	−0.301
**NEFA [μM]**	−0.1	0.124	0.025	−0.014	−0.031	0	0.237	0.19	1	−0.564 *	−0.212	0.14	0.366	−0.153	−0.301	−0.320 *	−0.372 **	−0.340 *	−0.372 **	−0.293 *	−0.480 **	−0.226	0.035	−0.067	−0.205	0.036	0.133	−0.228	−0.1	0.006	−0.203	−0.304
**TH stain**	0.355	−0.19	−0.415	−0.181	0.227	0.498 *	−0.267	−0.336	−0.564 *	1	--	--	--	--	--	0.713 **	0.456	0.641 **	0.545 *	0.709 **	0.482	−0.198	0.007	0.286	−0.243	−0.13	0.052	0.703	0.002	−0.16	0.218	0.004
**NAc *Oprm1***	0.431 *	−0.208	−0.254	−0.730 **	0.131	0.254	−0.286	−0.331	−0.212	--	1	0.537 **	0.525 **	0.560 **	0.657 **	−0.102	−0.073	−0.072	−0.064	−0.16	−0.153	0.419	−0.105	−0.394	−0.151	−0.724 **	--	--	0.315	0.694	0.447	0.857
**NAc *Per3***	−0.01	0.051	−0.163	−0.425	0.323	0.141	−0.215	−0.009	0.14	--	0.537 **	1	0.547 **	0.231	0.346	0.498	0.489	0.494	0.514	0.444	0.41	−0.026	−0.132	0.054	−0.106	−0.458	--	--	0.133	0.961	0.268	--
**NAc *Pts***	0.041	−0.234	−0.437 *	−0.091	−0.103	−0.125	−0.249	−0.066	0.366	--	0.525 **	0.547 **	1	0.333	0.292	−0.538	−0.468	−0.438	−0.471	−0.654 *	−0.660 *	0.225	0.281	0.119	0.058	0.168	--	--	0.574	0.217	0.792	0.463
**NAc *Slc6a3***	0.383 *	−0.228	−0.17	−0.594 *	0.085	0.262	−0.22	−0.303	−0.153	--	0.560 **	0.231	0.333	1	0.431 *	−0.221	−0.19	−0.206	−0.204	−0.217	−0.183	0.109	−0.228	−0.321	−0.3	−0.443	--	--	0.838	−0.853	0.779	−0.688
**NAc *Esr2***	0.354	−0.124	0.139	−0.666 *	−0.046	0.057	−0.235	−0.305	−0.301	--	0.657 **	0.346	0.292	0.431 *	1	−0.2	−0.195	−0.183	−0.193	−0.19	−0.197	0.129	−0.327	−0.331	−0.482	−0.476	--	--	0.902	−0.295	0.980 *	−0.037
**BAT *Ucp1***	−0.144	0.091	0.023	0.098	0.097	0.069	−0.062	−0.047	−0.320 *	0.713 **	−0.102	0.498	−0.538	−0.221	−0.2	1	0.786 **	0.755 **	0.536 **	0.721 **	0.812 **	0.059	0.179	0.336 *	0.255	0.229	−0.211	−0.183	−0.262	−0.162	−0.156	0.15
**BAT *Adrb3***	0.132	−0.183	−0.198	−0.066	0.066	0.141	−0.284 *	−0.264	−0.372 **	0.456	−0.073	0.489	−0.468	−0.19	−0.195	0.786 **	1	0.875 **	0.612 **	0.740 **	0.866 **	0.053	0.219	0.242	0.306 *	0.021	−0.137	0.127	−0.098	0.14	0.028	0.197
**BAT *Esr1***	0.101	−0.144	−0.235	−0.102	0.065	0.202	−0.226	−0.256	−0.340 *	0.641 **	−0.072	0.494	−0.438	−0.206	−0.183	0.755 **	0.875 **	1	0.662 **	0.770 **	0.772 **	0.096	0.225	0.321 *	0.394 **	0.053	−0.122	0.214	0.019	0.165	0.165	0.414 *
**BAT *Esr2***	0.203	−0.199	−0.213	−0.246	0.166	0.244	−0.204	−0.275 *	−0.372 **	0.545 *	−0.064	0.514	−0.471	−0.204	−0.193	0.536 **	0.612 **	0.662 **	1	0.489 **	0.830 **	−0.077	−0.007	0.112	0.114	−0.005	−0.112	0.057	0.063	0.201	0.314	0.274
**BAT *Adipoq***	0.138	−0.148	−0.21	−0.01	0.151	0.189	−0.238	−0.264	−0.293 *	0.709 **	−0.16	0.444	−0.654 *	−0.217	−0.19	0.721 **	0.740 **	0.770 **	0.489 **	1	0.600 **	−0.094	0.147	0.267	0.103	−0.06	−0.07	0.293	−0.076	0.078	0.126	−0.035
**BAT *Lept***	0.157	−0.207	−0.232	−0.207	0.325 *	0.197	−0.315 *	−0.317 *	−0.480 **	0.482	−0.153	0.41	−0.660 *	−0.183	−0.197	0.812 **	0.866 **	0.772 **	0.830 **	0.600 **	1	−0.191	−0.135	0.05	0.087	−0.166	−0.112	0.08	−0.017	0.113	0.122	0.426 *
**PGAT *Ucp1***	−0.01	−0.046	−0.153	−0.051	−0.024	0.144	−0.087	−0.099	−0.226	−0.198	0.419	−0.026	0.225	0.109	0.129	0.059	0.053	0.096	−0.077	−0.094	−0.191	1	0.269	0.09	0.341 *	0.116	−0.076	−0.119	0.307	−0.055	0.682 **	−0.157
**PGAT *Adrb3***	0.189	−0.196	−0.232	0.146	−0.215	0.024	−0.243	−0.266	0.035	0.007	−0.105	−0.132	0.281	−0.228	−0.327	0.179	0.219	0.225	−0.007	0.147	−0.135	0.269	1	0.636 **	0.557 **	0.324 *	0.193	−0.297	−0.05	0.101	−0.012	−0.073
**PGAT *Esr2***	0.231	−0.258	−0.264	−0.052	−0.233	−0.077	−0.249	−0.319 *	−0.067	0.286	−0.394	0.054	0.119	−0.321	−0.331	0.336 *	0.242	0.321 *	0.112	0.267	0.05	0.09	0.636 **	1	0.604 **	0.186	0.009	−0.164	0.049	−0.026	−0.059	0.226
**PGAT *Adipoq***	0.154	−0.161	−0.236	−0.084	−0.175	−0.044	−0.279 *	−0.218	−0.205	−0.243	−0.151	−0.106	0.058	−0.3	−0.482	0.255	0.306 *	0.394 **	0.114	0.103	0.087	0.341 *	0.557 **	0.604 **	1	0.255	0.041	−0.373	0.134	−0.107	0.148	0.410 *
**PGAT *Lept***	−0.528 **	0.652 **	0.471 **	0.543 **	−0.251	−0.375 **	0.439 **	0.521 **	0.036	−0.13	−0.724 **	−0.458	0.168	−0.443	−0.476	0.229	0.021	0.053	−0.005	−0.06	−0.166	0.116	0.324 *	0.186	0.255	1	−0.13	−0.648 **	−0.244	−0.316	−0.105	−0.305
**BAT UCP1 stain**	0.31	−0.204	−0.246	−0.318	0.16	0.236	0.022	−0.118	0.133	0.052	--	--	--	--	--	−0.211	−0.137	−0.122	−0.112	−0.07	−0.112	−0.076	0.193	0.009	0.041	−0.13	1	0.234	0.283	0.09	0.308	0.265
**PGAT UCP1 stain**	0.38	−0.382	−0.423	−0.697 **	0.122	0.543 *	−0.332	−0.540 *	−0.228	0.703	--	--	--	--	--	−0.183	0.127	0.214	0.057	0.293	0.08	−0.119	−0.297	−0.164	−0.373	−0.648 **	0.234	1	0.199	0.027	0.024	0.197
**PGAT ESR1**	0.451 *	−0.285	−0.435 *	−0.385 *	0.467 **	0.324	−0.333	−0.396 *	−0.1	0.002	0.315	0.133	0.574	0.838	0.902	−0.262	−0.098	0.019	0.063	−0.076	−0.017	0.307	−0.05	0.049	0.134	−0.244	0.283	0.199	1	0.668 **	0.569 **	0.2
**PGAT ESR2**	0.547 **	−0.400 *	−0.356	−0.388 *	0.448 *	0.351	−0.35	−0.377 *	0.006	−0.16	0.694	0.961	0.217	−0.853	−0.295	−0.162	0.14	0.165	0.201	0.078	0.113	−0.055	0.101	−0.026	−0.107	−0.316	0.09	0.027	0.668 **	1	0.324	0.15
**PGAT UCP1**	0.356 *	−0.191	−0.421 *	−0.376 *	0.314	0.379 *	−0.236	−0.343	−0.203	0.218	0.447	0.268	0.792	0.779	0.980 *	−0.156	0.028	0.165	0.314	0.126	0.122	0.682 **	−0.012	−0.059	0.148	−0.105	0.308	0.024	0.569 **	0.324	1	−0.009
**PGAT ADIPOQ**	0.176	−0.31	−0.343	−0.331	0.3	0.28	−0.376 *	−0.301	−0.304	0.004	0.857	--	0.463	−0.688	−0.037	0.15	0.197	0.414 *	0.274	−0.035	0.426 *	−0.157	−0.073	0.226	0.410 *	−0.305	0.265	0.197	0.2	0.15	−0.009	1

Pearson’s bivariate correlations reported as r values; * *p* < 0.05; ** *p* < 0.01; n’s ranged from 17 to 69, dependent upon available data.

**Table 2 ijms-25-06130-t002:** Primers used for rtPCR.

Gene ID	Company	Forward	Reverse	Product Size
*18s*	IDT (Coralville, IA)	TCAAGAACGAAAGTCGGAGG	GGACATCTAAGGGCATCAC	488
*PTS*	Sigma (Aizu, Japan)	CGATGAAGAGAACTTAAGAGTG	TGTAACAGGATCAATCTCTCC	106
*Per3*	Sigma	GAGAGTATGTCATTCTGGATTC	TCATTTAATGGACTCGTTCG	109
*DRD1*	IDT	TCTCCCAGATCGGGCATTT	GGGCCTCTTCCTGGTCAATC	131
*DRD2*	Sigma	TTGTTCTTGGTGTGTTCATC	TATAGATGATGGGGTTCACG	150
*OPRM1*	Sigma	ACTCATGTTGAAAAACCCTG	CTGTGTTCAGATGACATTCAC	125
*Slc6a3*	Sigma	CCAATAACTGCTATAGAGATGC	GTAGATGATGAAGATCAACCC	167
*Esr1*	Sigma	CAAAGGTAAATGTGTGGAAGG	GTGTACACTCCGGAATTAAG	141
*Esr2*	Sigma	CTCAACTCCAGTATGTACCC	CATGAGAAAGAAGCATCAGG	178
*Cyp19a1*	Sigma	AACATCATTCTGAACATCGG	AGGGAACATTCTTCTCAAAG	94
*Adipoq*	IDT	GCACTGGCAAGTTCTACTGCAA	GTAGGTGAAGAGAACGGCCTTGT	
*Leptin*	IDT	CCTATTGATGGGTCTGCCCA	TGAGCGCTACCTGCATAGAC	
*Ucp1*	IDT	CACGGGGACCTACAATGCTT	ACAGTAAATGGCAGGGGACG	
*Adrb3*	IDT	AGGACCTGACCCTGTCATCC	TCTAAGCCTTTCATGCCCACA	

## Data Availability

All raw data are available upon request.

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
