# Peer review of "Knockdown of Esr1 from DRD1-Rich Brain Regions Affects Adipose Tissue Metabolism: Potential Crosstalk between Nucleus Accumbens and Adipose Tissue"

_ijms, 2024, doi:10.3390/ijms25116130_

Round 1

Reviewer 1 Report

Comments and Suggestions for Authors

The manuscript evaluated KD of Esr1 and how this affects adipose tissue. The study is interesting but the possible mechanisms are missing. Also, the correlation is not straightfoward and methods are not well-described which must be addressed for proper appreciation.

Introduction

The introduction is fine and authors bring the hypothesis of the study which is crucial.

Results

- Authors should insert a rationale at the begining of each paragraph. This would provide better understanding for the reader concerning the purpose of the experiment performed.

- There are several results described as different even though the statistics between groups were not. This may raise questions whether the conclusion are straighfoward. I would recommend to emphasize only the statistically distinct results. Also, remove p value from all statistically different results to make the text more clearer.

-Why did authors choose animals at 23 weeks old ? The hormonal millieu changes along life, specially in females. 

- Did autors check RNA quality by agarose gel or other methods ?

- In Figure 3, authors described lipid droplet depletion based on nuclei density and histology, but this is not convincing, mainly because any stattistical difference was found. I would recommend more suitable staining such as Oil Red O or Sudan Black for confirmation. Also, B panel in this Figure is too small and the reader can not get any information from the images. Please, address. 

- Figure 3.D. Please, insert proper gene names.

- BAT activation would be confirmed by in vitro assays in modifed adipocytes under OROBOROS or Seahorse experiments. The current data is not convincing to sustain this statement.

- Figure 5. A - How did authors normalized pixel intensity ? Is it number of pixels, instead ? C. Genes should be written italicized for mice. D - Insert western blotting bands with loading control for better appreciation.

- Figure 6.A How was FI normalized ? Per um² ? Or just ramdom fields ? Please, clarify in the Y axis. Image magnification is missing. Please, insert scale bar and magnification in the legend.

- There are few statistically different results, specially in the biometric data. Could this be due to increased expression of Esr2 as a compensation ?

- It is not clear to this reviewer the underlying mechanism (or potential mechanism) whereby ESR1 depletion would affect adipocytes. A summary figure would assist with this issue.

- Authors studied male mice but did not discuss the difference compared to female based on sexual hormones. How androgens could affect the results ?

- Why males had increased insulin after fasting (considering the trend) ?

- Histology images are missing scale bars and maginification details in the legend. Also, were section stained with hematoxilyn for nuclei visualization ?

- How could KD of Esr1 in brain affect Esr2 in BAT ? This is not clear to me.

- Authors stated in the limitations and in the title that their experimental approach are closely to a knock-down rather knockout. Therefore, the KO should be removed from the figures and stated properly.

- Validation of the statistically different genes by western blotting would strenght the conclusions, such as for Oprm.

Author Response

Thank you so much for your careful read of our study and for your excellent suggestions for improvement. We have addressed all concerns where possible and we think the manuscript is improved after these modifications. 

Add rationale to beginning of each paragraph of the results section

Thank you for this suggestion; we have added a short rationale for each experiment, where such rationale was not already present (e.g., sections 4, 6, and 8).

Remove p vales that were significant; only indicate those that were not, and refrain from indicating significance when not significant.

We have edited the manuscript to better clarify statistical significance and have removed all significant p vales from the text; we now only show trending p values to illustrate specific points.

-Why did authors choose animals at 23 weeks old ? The hormonal millieu changes along life, specially in females

Thank you for this question – while many studies are conducted on younger mice, we wanted to ensure we had sufficient time to assess phenotypic differences well after sexual maturity (which occurs @~12 weeks)

- Did autors check RNA quality by agarose gel or other methods ?

Yes, we test each sample’s RNA quality using the Nanodrop. All samples had 260/280 values between 1.9-2.0, indicating pure RNA free of contamination.

- In Figure 3, authors described lipid droplet depletion based on nuclei density and histology, but this is not convincing, mainly because any stattistical difference was found. I would recommend more suitable staining such as Oil Red O or Sudan Black for confirmation.

The lipid droplet depletion, we found to be striking, but unfortunately did not have the capacity to perform Oil Red O staining to objectively quantify lipid droplets. We used nuclei density and histology because these were additional assessments we could make on the stained samples. Although we did not detect statistical significance, likely due to underpowering of only n=5/group, we thought the reader would appreciate seeing the data, despite not reaching statistical significance. In general, adipose tissue browning is assessed subjectively based on subjective characteristics such as the multilocular phenotype and lipid content. The objective measure is UCP1 content, which we quantified via staining, as well as western blot and rtPCR.

Also, B panel in this Figure is too small and the reader can not get any information from the images. Please, address. 

We increased the size of those representative images – thank you.

- Figure 3.D. Please, insert proper gene names.

We inserted proper gene names in Figure 3D.

 BAT activation would be confirmed by in vitro assays in modifed adipocytes under OROBOROS or Seahorse experiments. The current data is not convincing to sustain this statement.

This is an excellent suggestion that we will implement in future experiments. Unfortunately, those ex vivo assays need to be done with fresh tissue, which cannot be performed for this particular study because the study has already been completed. We now soften the language regarding BAT activation and indicate this as a limitation and opportunity for future work.

- Figure 5. A - How did authors normalized pixel intensity ? Is it number of pixels, instead ?

Thank you for this opportunity to clarify. Integrated intensity is the sum of the pixel intensity over all of the pixels in the image. For clarification, we have changed “pixels” to “mean stain intensity”. 

  1. Genes should be written italicized for mice.

Corrected – thank you!

D - Insert western blotting bands with loading control for better appreciation.

We have included representative images and indicate where the band appeared. We can also provide all raw data including full blots if necessary. Thank you.

- Figure 6.A How was FI normalized ? Per um² ? Or just ramdom fields ? Please, clarify in the Y axis. Image magnification is missing. Please, insert scale bar and magnification in the legend.

We changed the Y axis to reflect the normalized units of “mean pixel density”, and indicated the magnification in the legend.  We also added scale bars to each image.

- There are few statistically different results, specially in the biometric data. Could this be due to increased expression of Esr2 as a compensation ?

Thank you for this comment. We agree that Esr2 compensation likely may have contributed to our lack of statistical differences in many cases. We had not make this hypothesis clear originally and have now added language regarding this to the discussion.

At a figure to depict the hypothetic mechanism

We liked this suggestion and have added a hypothetical model figure (Figure 7).

- Authors studied male mice but did not discuss the difference compared to female based on sexual hormones. How androgens could affect the results ?

This is a great question that we will address in future studies. We did not measure androgens in this study.

- Why males had increased insulin after fasting (considering the trend) ?

It is typical that males have greater insulin resistance than females. Higher levels of fasted insulin are indicative of this and this was an expected finding.

- Histology images are missing scale bars and maginification details in the legend. Also, were section stained with hematoxilyn for nuclei visualization ?

Scale bars and magnification details have been enlarged (they were present in the original, but too small). We have now added this information to the legend as well. Images were stained with UCP-1 Abs, but not H&E.

- How could KD of Esr1 in brain affect Esr2 in BAT ? This is not clear to me.

This is a great question! This connection between DRD1-selective Esr1 KD and Esr2 in BAT has not been demonstrated before, to our knowledge, and certainly requires further study. However, our previous studies demonstrate that activating brown and beige adipose tissues induces Esr2 in adipose tissue. We hypothesize that suppression of Esr1 in the brain caused an increase in ERb signaling, which is known to induce dopamine. Dopamine is a neurotransmitter important in activating BAT and browning of white adipose tissue, and it does this through its conversion to norepinephrine. Thus, we think the adipose tissue specific increase in ERb was an indirect effect of the brain-specific mutation (see Figure 7, a hypothetical model constructed upon receiving the excellent suggestion).

- Authors stated in the limitations and in the title that their experimental approach are closely to a knock-down rather knockout. Therefore, the KO should be removed from the figures and stated properly.

We have edited the paper throughout to change KO to KD. Thank you for the suggestion – it is indeed more appropriate to describe the mutated mouse as “KD” rather than KO.

- Validation of the statistically different genes by western blotting would strenght the conclusions, such as for Oprm.

Unfortunately, we used all of the NAc tissue and cannot conduct additional analyses. We added lack of protein expression validation as a limitation.

Reviewer 2 Report

Comments and Suggestions for Authors

Very interesting work. Estrogens play animportant role in our body. Any change inthe level of estrogenscauses all kinds ofchanges taking place in the body. True, thestudies were conducted on mice, but it isknownthat the correlation of the studieswill be very similar. Therefore, it is worthinvestigating changes in estrogenanddopamine in humans and check themechanisms that occur and affect ourbody. A lot of work has alreadybeen doneon lifestyle and hormone correlation. However, the more research is done thatwill enable us tounderstand the cognitivefunctions of behavior and physical activity, the sooner we will be able tointroducehormonal "prophylaxis". In addition, thework used a wide range of studies, basedon a thoroughanalysis. As I mentionedearlier, I propose to prepare a similarresearch program for patients and to verifyandcompare the results obtained.
How estrogen loss exactly affectsmetabolic health, leading to increasedobesity, insulin resistanceandcardiometabolic diseases, however, themechanisms are not fully understood. Wealso know that the loss ofestrogen mainlyoccurs during the menopausal period andfrom that moment begins the process ofmetallichormonal dysfunction. Knowingthe model of estrogen change, this slingcould be slowed down, which wouldlead toslowing down the formation of, amongothers, the above-mentioned diseaseunits.
Literature, not extensive. Table 1 is hard toread. It is worth tempting to separatetables.

Author Response

Thank you for acknowledging the importance of this preclinical work and the critical importance of translation to humans to prevent metabolic dysfunction and behavioral changes following menopause. Your positive comments are appreciated, as is the time you invested to read our work. We have improved the clarity of Table 1 by enlarging the tables and showing them landscape to improve readability – thank you for the suggestion.

Reviewer 3 Report

Comments and Suggestions for Authors

In the paper entitled “Knockdown of Esr1 from DRD1-rich brain regions affects adipose tissue metabolism: Potential crosstalk between nucleus accumbens and adipose tissueDusti Shay and collegues investigated the potential biological role of loss of signaling through estrogen receptor 1 and dopamine receptor in brain accumbens in association with a wide-range of behavioral an methabolic pathologies. For the purpose Authors created a Cre-lox knockout mouse model DRD1ERKO lacking DA receptor 1and Estrogen receptor. The validated model was studied for body weight and composition, energy intake and expenditure, insulin sensitivity and glucose tolerance, the effects on BAT and WAT UCP1 expression and Tyrosin Hydroxylase content to correlate with basal metabolism. Authors concluded that its DRD1ERKO model showed some limits not affecting spontaneous physical activity or adiposity, as they supposed, the mutation led to sex-divergent changes in adipose tissues biology. Females showed an increase in BAT activity and browning of WAT and improve glucose tolerance, among males that showed an increase in WAT leptin. Altogether, Authors shoed a novel relationship between estrogen signaling and DA-rich brain region, BAT activity and glucose homeostasis.

The paper is well written, the rationale is clear, and figures are exhaustive. Nevertheless, despite the limits of the new mice model, well been metabolically characterized, and the lack of protein expression data, Authors described that E2 signaling in the nucleus accumbens (NAc), the brain’s reward center, affects adipose tissue and systemic metabolism primally in female mice DRD1ERKO model.

Some concerns are to be assessed to the Authors:

-       Please, show up the declaration of Ethical Committee Approval for the study design utilizing mice model.

-       Please, rephrase line 76 of Introduction.

Comments on the Quality of English Language

The quality of English language is good and fluent.

Author Response

Thank you for taking the time and effort to review our work, and for the comprehensive summary and positive comments. Thank you for pointing out the oversight regarding the addition of the ACUC approval statement. That has now been added. In addition, we revised sentence 76 of the introduction, as suggested. The sentence now reads: “Hence, the purpose of this investigation was to investigate how Esr1 knock-down on the DA receptor (DRD1) promotor (i.e., to target DA-rich brain regions, including the NAc) affects physical activity as well as systemic and adipose tissue metabolism.”

Round 2

Reviewer 3 Report

Comments and Suggestions for Authors

In the reviewed version of the paper entitled "Knockdown of Esr1 from DRD1-rich brain regions affects adi- 2 pose tissue metabolism: Potential crosstalk between nucleus ac- 3 cumbens and adipose tissue" Authors must be added the approval cod number of the IACUC ( line 519-520), this is mandatory. 

Author Response

Thank you so much for pointing out this oversight! I'm embarrassed that this was missing. The approval number and data have now been added to the manuscript (ACUC 9474, 11/9/2018) - I will upload then once I hear back from reviewers 1 and 2. Thank you again!